# TUNING LAYERNORM IN ATTENTION: TOWARDS EFFICIENT MULTI-MODAL LLM FINETUNING

**Bingchen Zhao*[1]    Haoqin Tu*[2]    Chen Wei[3]    Jieru Mei[3]    Cihang Xie[4]**

*equal contribution

[1] University of Edinburgh    [2] University of Chinese Academy of Sciences
[3] Johns Hopkins University    [4] UC Santa Cruz

## ABSTRACT

This paper introduces an efficient strategy to transform Large Language Models (LLMs) into Multi-Modal Large Language Models. By conceptualizing this transformation as a domain adaptation process, *i.e.*, transitioning from text understanding to embracing multiple modalities, we intriguingly note that, within each attention block, tuning LayerNorm suffices to yield strong performance. Moreover, when benchmarked against other tuning approaches like full parameter finetuning or LoRA, its benefits on efficiency are substantial. For example, when compared to LoRA on a 13B model scale, performance can be enhanced by an average of over 20% across five multi-modal tasks, and meanwhile, results in a significant reduction of trainable parameters by 41.9% and a decrease in GPU memory usage by 17.6%. On top of this LayerNorm strategy, we showcase that selectively tuning only with conversational data can improve efficiency further. Beyond these empirical outcomes, we provide a comprehensive analysis to explore the role of LayerNorm in adapting LLMs to the multi-modal domain and improving the expressive power of the model.

## 1 INTRODUCTION

Large Language Models (LLMs) have had many application scenarios since their debut. In particular, extending LLMs to handle multiple modalities has gathered much interest from both academia and industry. Such models, termed *Multi-modal* Large Language Models (MLLMs), are typically derived by finetuning a pretrained LLM on multi-modal data (Liu et al., 2023; Ye et al., 2023). However, this process typically poses a substantial computational challenge (Liu et al., 2023), particularly for exceptionally large-scale models. While Su et al. (2023); Zhang et al. (2023) employ low-rank adapters (LoRA) (Hu et al., 2022) or soft prompts (Li & Liang, 2021a) for more parameter-efficient tuning, this often comes at the cost of compromised performance on multi-modal tasks. This challenge prompts the pivotal question: *how can we make this process more efficient?*

In response to this challenge, we introduce a simple and effective strategy for MLLM finetuning: as illustrated in Figure 1(a), within each attention block, we adjust only the weights of the LayerNorm (Ba et al., 2016). This strategy is underpinned by the understanding that the evolution from LLMs to MLLMs can be conceptualized as a domain adaptation process, *i.e.*, transitioning from text-centric to multi-modal understanding. Adjusting normalization layers, as suggested by prior research, emerges as a particularly effective technique in such domain shifts (Li et al., 2016). Empirically, this straightforward technique can surprisingly yield comparable or even better performance than the strong baseline of finetuning all parameters offer about $10\times$ more parameter efficiency than LoRA.

By delving deeper, we note that the process can be further simplified by designating LayerNorm as the sole trainable component within the entire model. This means, in contrast to the typical configurations depicted in Figure 1(a)-(c), we now freeze the standardly activated elements, including the vision-language connector, word embedding, and the output head. We term it as LayerNorm-simple. Impressively, despite constituting a mere 0.004% of trainable parameters, this configuration surpasses the performance of LoRA, registering an average enhancement of 4.3% across five benchmarks.

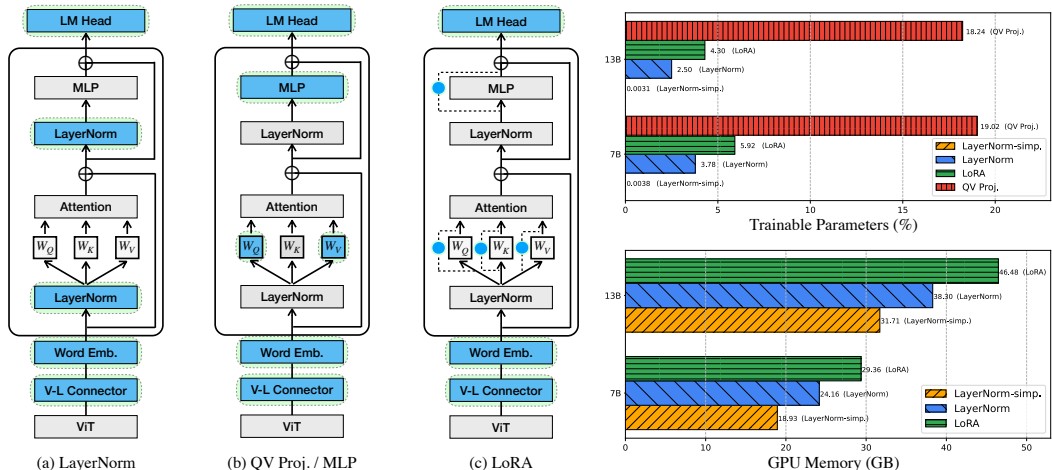

Figure 1: (*left*) Different tuning methods for MLLMs. Trainable components are in blue, while frozen parameters are in gray. Within the attention blocks, (*a*) only activates LayerNorm parameters. Note that vision-language connector, word embedding, and output head paramters are by default activated for all three options. (*right*) Comparison on trainable parameters and GPU memory. Tuning LayerNorm achieves significant reductions in trainable parameters and GPU memory usages.

On top of this LayerNorm strategy, we further improve the finetuning efficiency from the data perspective. Specifically, we assess the performance implications of different types of finetuning data, including conversational data, detailed description data, and complex reasoning data. Our results offer a crucial insight: not all data are created equal for the task of MLLM finetuning. Remarkably, we find that MLLMs finetuned on conversational data consistently outperform those finetuned on other data types. Specifically, conversational data improves the model performance by an average of 50% compared to other data types. This observation interestingly opens up avenues for more targeted data collection and curation strategies, thereby further optimizing the efficiency of MLLMs finetuning. Furthermore, by combining the LayerNorm strategy and this data perspective, we can achieve on average 10.0% performance improvement over full parameter finetuning on traditional VQA benchmarks with an LLAMA2 13B model while using significantly less parameters and data.

Beyond the empirical outcomes above, we conduct an investigation into the expressive power of LayerNorm tuning. Our analysis reveals that LayerNorm-tuned MLLMs exhibit lower cross-layer similarity compared to models all of which parameters are finetuned. This lowered similarity is indicative of a more expressive model, since the model incorporates anisotropic layer presentations can capture a wider range of learning patterns (Pires et al., 2023). It stands to reason that this amplified expressiveness is a key factor underpinning the efficiency and superior performance we noted, granting the model enhanced adaptability to novel multi-modal datasets.

In essence, our findings illuminate the profound influence of LayerNorm tuning, suggesting its potential to adeptly harness the intrinsic properties of LLMs. We hope that this study will catalyze subsequent research endeavors focused on efficient multi-modal finetuning.

## 2 RELATED WORKS

**Multi-Modal and Large Language Models.** Multi-modality has been extensively studied in the literature. Starting from learning aligned representation across image and text modalities like CLIP (Radford et al., 2021), many works have proposed more similar techniques (Yu et al., 2022; Mu et al., 2022; Zhao et al., 2023) and try to explain the internal mechanism of models using multiple and efficient representations (Chen et al., 2023; 2024). Given the interest in developing instruction-tuned LLMs (Ouyang et al., 2022), the studies of multi-modal models have also shifted the focus to instruction-tuned MLLMs. For example, LLAVA (Liu et al., 2023) pioneers the development of instruction-tuned MLLMs by designing instruction-tuning data with the help of GPT-4. The concurrent work MINIGPT4 (Zhu et al., 2023) is built using QFormers (Li et al., 2023a) and VICUNA (Zheng et al., 2023), but with only a linear layer activated for tuning. Similar to MINIGPT4,

Su et al. (2023) devises PANDAGPT with a more advanced vision encoder and LoRA-tuned LLM as its base models. MPLUG-OWL (Ye et al., 2023) mixes text-only and multi-modal instruction data for finetuning LLAMA model. INSTRUCTBLIP (Dai et al., 2023) builds on the BLIP2 model (Li et al., 2023a) with additional finetuning on instruction tuning datasets.

**Parameter-Efficient Finetuning.** The parameter-efficient finetuning (PEFT) technique has been widely applied and studied because of huge resource consumption of larger and larger deep learning models. Adapter (Houlsby et al., 2019) additionally adds trainable adapter components in the LM, which proved to achieve comparable results in NLP tasks with less than 10% trainable parameters. Prefix tuning (Li & Liang, 2021b) only inserts trainable parameters to the attention head in LMs LoRA (Hu et al., 2022), as the most widely employed PEFT method recently, injects trainable low rank decomposition matrices into a model to reduce the number of training parameters. Following the same line, QLoRA (Dettmers et al., 2023) achieves further reduction in the memory usage for finetuning LLMs with quantized 4-bits parameters. These techniques have been widely utilized for the tuning of LLMs (He et al., 2022; Tu et al., 2022), MLLMs (Zhu et al., 2023; Tu et al., 2023) and other applications (Dutt et al., 2023; Xu et al., 2023). In this paper, we show that tuning LayerNorm in the LLM of an MLLM can achieve better results than tuning other components in the model and LoRA tuning, while requiring less resource for computation.

**The Normalization Studies.** The normalization layer in neural networks has long been a subject for debate and study. The foundational work on batch normalization (Ioffe & Szegedy, 2015) first introduces normalization as an important part of neural network architectures, and argues the effectiveness of normalization comes from alleviating the internal covariant shifting problem. Later works have proposed many variants of normalization, such as InstanceNorm (Ulyanov et al., 2016), GroupNorm (Wu & He, 2018), and LayerNorm (Ba et al., 2016). LayerNorm has been the design choice of LLMs for normalization, and its effectiveness in LLM pretraining has also been explored and discussed (Xu et al., 2019). In this work, we explore the effectiveness of finetuning LayerNorm in MLLMs as well as the reason behind the effectiveness.

## 3 TUNING AND EVALUATION SETTINGS

In this section, we first introduce the comment structure of MLLMs, different tuning strategies of MLLMs, and then present the evaluation benchmarks employed in the paper.

**Architecture of MLLMs.** A typical MLLM usually contains three parts: 1) A vision encoder for extracting visual features; 2) An LLM decoder for generating plausible texts from both text instructions and visual information; 3) A vision-language connector for bridging between the vision encoder and the LLM. We follow Liu et al. (2023) to set up the model, where the vision encoder is a CLIP-pretrained ViT-L (Radford et al., 2021) and the vision-language connector is a simple linear projector.

We experiment with a range of LLMs for the language decoder. Specifically, we choose three types of LLM with 7B and 13B scales: VICUNA-7B (v1.1) (Zheng et al., 2023), LLAMA2-7B&13B, and LLAMA2-CHAT-7B&13B (Touvron et al., 2023).

**Baseline Models.** Apart from our trained MLLMs, we showcase the performances of three publicly available models: MPLUG-OWL (Ye et al., 2023), MINIGPT4 (Zhu et al., 2023), and LLAVA-V0 (Liu et al., 2023). The LLAVA-V0 here represents the initial release of LLAVA. These baseline results are obtained from existing literature (Fu et al., 2023; Li et al., 2023b) or tested using their released checkpoints.

**Tuning Modules.** To analyze the effects of different tuning components in the MLLMs, we employ five different tuning paradigms on the same training corpus. (1) finetune: activates all the parameters in LLM for MLLMs tuning; (2) LoRA: inserts LoRA component (Hu et al., 2022) with rank 32 between all linear structure in the LLM; (3) Attn. QV Proj.: activates Q and V linear projection in attention of the LLM, as they are proved to be especially effective for tuning LLMs (Hu et al., 2022); (4) Attn. MLP: activates MLP layers in attention of the LLM; (5) LayerNorm: both input and post LayerNorm in attention blocks of the LLM. Note that all tuning methods activate vision-language connector for training.

Table 1: Model performance on five multi-modal benchmarks with different components tuned in the LLM. We mark the best results with **bold** and the second best scores with underline. '-' means the model cannot follow the required output format on captioning tasks.

| Models | MME↑ | VQAv2↑ | MSCOCO↑ | Flickr30k↑ | POPE↑ |
|---|---|---|---|---|---|
| BASELINE MODELS | | | | | |
| MPLUG-OWL | 967.4/276.1 | 21.38 | 70.70 | 41.78 | 50.9/54.0/50.7 |
| MINIGPT4 | 866.6/292.1 | 17.30 | - | - | 69.7/79.7/65.2 |
| LLAVA-v0 | 502.8/214.6 | 15.06 | 58.89 | 23.02 | 70.5/74.6/66.0 |
| MM-VICUNA-7B | | | | | |
| Finetune | 625.2/270.7 | 15.40 | 67.50 | 34.61 | 73.8/76.5/66.5 |
| LoRA | 552.3/217.5 | 15.00 | 63.93 | 34.13 | 50.4/51.6/50.4 |
| Attn. QV Proj. | 678.0/**277.5** | 15.51 | 72.63 | 32.24 | 72.0/77.1/65.3 |
| Attn. MLP | 637.3/268.2 | 15.37 | 65.22 | 37.47 | 60.0/68.2/56.6 |
| LayerNorm | **723.2**/253.2 | 17.06 | **80.89** | **48.01** | **76.1/81.1/70.8** |
| LayerNorm-simp. | 720.9/251.8 | **23.46** | 79.75 | 46.18 | 61.1/72.3/58.5 |
| MM-LLAMA2-7B | | | | | |
| Finetune | **661.3/237.1** | 16.09 | 65.08 | 31.64 | 56.3/65.0/55.4 |
| LoRA | 395.0/200.0 | 14.87 | 61.97 | 26.17 | 51.9/54.7/51.3 |
| Attn. QV Proj. | 584.0/222.9 | 16.39 | 76.05 | 42.93 | 55.7/63.0/56.8 |
| Attn. MLP | 413.1/203.6 | 15.29 | 58.35 | 29.04 | 53.7/59.6/53.9 |
| LayerNorm | 583.2/200.7 | **16.78** | **88.85** | **49.24** | **66.6/68.5/64.9** |
| LayerNorm-simp. | 542.6/205.0 | 14.98 | 65.10 | 46.88 | 51.6/52.5/51.1 |
| MM-LLAMA2-CHAT-7B | | | | | |
| Finetune | 805.4/234.6 | 15.29 | 66.33 | 26.70 | 60.3/69.8/57.9 |
| LoRA | 709.8/228.6 | 15.28 | 57.27 | 25.49 | 59.2/65.9/56.8 |
| Attn. QV Proj. | **926.5**/220.7 | 15.88 | 58.49 | 31.10 | 68.5/77.3/65.0 |
| Attn. MLP | 840.0/**240.0** | 15.20 | 54.42 | 24.89 | 56.9/67.3/56.8 |
| LayerNorm | 651.3/219.3 | 16.60 | **75.34** | **43.75** | **71.3/72.4/67.8** |
| LayerNorm-simp. | 372.0/169.3 | **18.42** | 59.99 | 41.63 | 52.0/54.6/52.3 |
| MM-LLAMA2-13B | | | | | |
| Finetune | 402.3/199.3 | 18.33 | 73.88 | 45.33 | 51.6/51.1/52.2 |
| LoRA | 400.8/189.3 | 16.08 | 68.83 | 43.70 | 50.5/51.2/50.5 |
| Attn. QV Proj. | 489.0/**200.4** | 15.12 | 63.07 | 32.81 | 51.1/52.9/52.5 |
| Attn. MLP | 387.5/167.1 | **25.19** | 64.19 | 44.06 | 50.7/52.2/51.9 |
| LayerNorm | **526.0**/177.5 | 15.31 | **82.92** | **48.42** | **60.0/69.1/58.9** |
| LayerNorm-simp. | 403.3/185.4 | 18.62 | 68.28 | 43.04 | 55.3/58.0/57.2 |
| MM-LLAMA2-CHAT-13B | | | | | |
| Finetune | 623.3/221.4 | 15.17 | 64.19 | 41.82 | 67.6/64.8/64.5 |
| LoRA | 516.7/214.3 | 14.39 | 66.33 | **43.09** | 66.9/64.1/63.8 |
| Attn. QV Proj. | 624.5/250.4 | 14.91 | 60.96 | 34.90 | 66.3/66.0/61.8 |
| Attn. MLP | 456.7/211.4 | 14.67 | 62.19 | 40.39 | 56.8/56.9/56.5 |
| LayerNorm | **929.3/254.3** | **16.10** | **74.96** | 42.79 | **78.9/83.9/74.3** |
| LayerNorm-simp. | 824.3/221.1 | 13.29 | 52.70 | 40.20 | 73.3/76.0/69.0 |

**Training Details.** We pre-train the vision-language connector for 3 epochs on CC3M (Changpinyo et al., 2021), and conduct the finetuning stage on 80K filtered image-text pairs collected by Liu et al. (2023) for 1 epoch. For the first stage, we set the learning rate to 2e-3 for all variants. During the second stage, we search the learning rate from 2e-3 to 1e-7 with 11 options for all tuning strategies and pick the best learning rate based on their performances on Flickr30k task. We set the weight decay (Loshchilov & Hutter, 2019) to 0 and a warmup ratio to 0.03 with the cosine learning rate scheduler (Loshchilov & Hutter, 2017). Moreover, we employ the gradient checkpointing (Chen et al., 2016), DeepSpeed technique (Rajbhandari et al., 2020), and a data precision of TensorFloat32 for models training. We conduct all of our experiments on 4 80G A100 GPUs on the same node.

Table 2: Memory consumption and percentages of trainable parameters tested on a single A100 GPU.

| Model | 7B SCALE | | 13B SCALE | |
|---|---|---|---|---|
| | Mem. (GB) | #param. | Mem. (GB) | #param. |
| Finetune | OOM | 95.70% | OOM | 97.72% |
| LoRA | 29.4 | 5.92% | 46.5 | 4.30% |
| Attn. QV Proj. | 57.0 | 19.02% | OOM | 18.24% |
| Attn. MLP | OOM | 65.21% | OOM | 66.24% |
| LayerNorm | 24.2 | 3.78% | 38.3 | 2.50% |
| LayerNorm-simp. | 18.9 | 0.004% | 31.7 | 0.003% |

**Multi-Modal Benchmarks.** We test the visual-instruction tuned models on recent multi-modal evaluation benchmarks, where five multi-modal benchmarks are deployed: MME (Fu et al., 2023) consists of two evaluation aspects, *i.e.*, cognition (CS) and perception (PS) with total 14 VQA tasks; VQAv2 (Antol et al., 2015), MSCOCO (Lin et al., 2014) and Flickr30k (Young et al., 2014) captioning tasks are commonly used benchmarks in the field of VQA and captioning. The former two benchmarks are based on MSCOCO-2017 dataset (Lin et al., 2014). For the latter two captioning tasks, we report the zero-shot CIDEr (Vedantam et al., 2015) scores (with three text-only QA examples) on the test set from Karpathy & Fei-Fei (2015). POPE (Li et al., 2023b) is used to evaluate the level of object hallucinations in MLLMs, which consists of three versions of balanced yes/no VQA tasks (*i.e.*, Popular/Random/Adversarial) considering objects in the given image.

# 4 EXPERIMENTAL RESULTS

## 4.1 TUNING LAYERNORM

**Tuning LayerNorm in Attention Blocks.** In table 1, it is noteworthy that activating only the LayerNorm yields the least activated parameters, yet the model performances are surprisingly impressive when compared to tuning other modules. Specifically, in two captioning tasks, the VQAv2 task, and the challenging hallucination benchmark POPE, models with only the LayerNorm activated consistently outperform all other competitors by at least 8.2%. On the comprehensively evaluated benchmark MME, while tuning LayerNorm outperforms finetuning the intact language model by an average of 6.6% on the Perception aspect, it lags behind finetuning by an average of 6.3% on the Cognition score. It is vital to note, however, that the LayerNorm only accounts for approximately 2.5% of the training parameters in the whole model.

In addition to tuning modules, another observation is that MLLMs incorporating human-aligned LLMs (such as LLAMA2-CHAT) exhibit superior performance in complex and demanding tasks such as POPE and MME compared to their unaligned counterparts. This underscores the importance of utilizing aligned LLMs to construct a more powerful MLLMs.

**Tuning LayerNorm and Only LayerNorm.** As the above LayerNorm method finetunes (1) vision-language connector, (2) word embedding, (3) output head, and (4) LayerNorm component in the LLM simultaneously, a pertinent question arises: *Is it possible for (4) LayerNorm alone to generalize effectively in training MLLMs?* To address this query, we take a step further and solely finetune LayerNorm in MLLMs, which is denoted as LayerNorm-simp. in table 1. The results are intriguing, demonstrating that even with a mere 0.004% parameter finetuning in the whole model, LayerNorm-simp. surpasses full parameter finetuning on three conventional vision-language tasks (i.e., two captioning and one VQA tasks) by 10%, and only lags behind full finetuning by 7.9% on the MME benchmark. This intriguing discovery suggests that the transition from LLM to MLLMs probably involves a domain adaptation process as the LayerNorm takes the most credits in tuning a well-behaved MLLMs. The LayerNorm alone may be also capable of integrating vision information with language tokens seamlessly.

**Memory Consumption and Parameter Efficiency.** In table 2, we present the total memory consumption and the percentage of trainable parameters of each MLLMs finetuning method across 7B and 13B scales. Methods like full parameter finetuning and finetuning MLPs in attention modules face out-of-memory (OOM) issue even on a high-capacity 80GB A100 GPU, while LayerNorm based

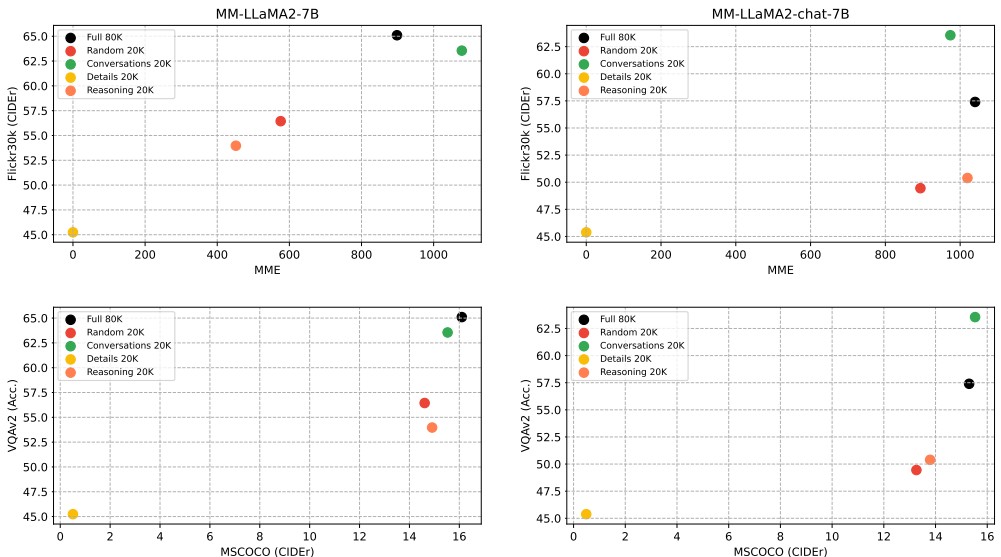

Figure 2: Performances of models that are finetuned on different datasets on four multi-modal benchmarks. The MME score is the sum of both Cognition and Perception scores on the benchmark.

methods stand out for their efficiency. Specifically, LayerNorm tuning requires only 24.2 GB and 38.3 GB memory at 7B and 13B scales respectively. Remarkably, LayerNorm-simp. further reduces the memory to 18.9 GB and 31.7 GB. In terms of trainable parameters, LayerNorm based methods also show remarkable efficiency, LayerNorm utilizes only 3.78% and 2.50% of the total parameters at the 7B and 13B scales, and LayerNorm-simp. takes efficiency to an extreme, involving only 0.004% and 0.003% of the parameters at these scales. These results demonstrate the efficiency advantage of LayerNorm tuning, compared with existing methods like LoRA or full parameter finetuning.

## 4.2 'LESS IS MORE' ON BOTH DATA AND PARAMETER SIDES

Efficiency in training can also be improved by considering the data used in LLMs and MLLMs (Zhou et al., 2023; Wei et al., 2023). To this end, we conducted experiments using LLAMA2-7B and LLAMA2-7B-CHAT, where we divided the training data into three categories, each comprising 20K data points: image-grounded conversation, image detail descriptions, and image-based complex reasoning, as previously deployed in Liu et al. (2023). Based on the results presented in fig. 2, we observe that the image-grounded conversation data is the most effective in enhancing the multi-modal capabilities of the model, with an average improvement of over 50% compared to other data types. This highlights the potential benefits of a targeted approach that leverages the strengths of specific data types to facilitate more nuanced and effective multi-modal tuning for MLLMs.

To validate 'Less is More' on both the data and parameter sides, we present results of MLLMs with LayerNorm activated in LLM and tuned on 20k conversational data in table 3. Our experimental results indicate that even with a smaller dataset and the use of LayerNorm tuning, the model outperforms the full parameter finetuning approach on the full 80K dataset by 18.4% on two captioning tasks, and only falls short in MME by a tolerable 2.5%. It is noteworthy that LayerNorm with 20K data is only 7.6% and 7.4% behind LayerNorm on the full 80K dataset for two captioning tasks and MME task, respectively. These findings demonstrate that 'Less is More' for both the parameter and data perspectives beyond language domain Zhou et al. (2023), but for multi-modal tuning.

## 5 INTUITIONS BEHIND LAYERNORM TUNING

In this section, driven by the empirical success of LayerNorm tuning, we explore the intuitions behind LayerNorm from three perspectives, domain adaptation, expressive power, and gradient variance.

Table 3: Model performance on different data types. Methods with 80K and Conv.20K suffix are tuned on the full 80K data and the 20K conversational data, respectively.

| Method | MME | VQAv2 | MSCOCO | Flickr30k | POPE |
|--------|-----|-------|--------|-----------|------|
| MM-VICUNA-7B | | | | | |
| Finetune-80K | 625.2/**270.7** | 15.40 | 67.50 | 34.61 | 73.8/76.5/66.5 |
| LayerNorm-80K | 723.2/253.2 | **17.06** | **80.89** | **48.01** | **76.1/81.1/70.8** |
| LayerNorm-Conv. 20K | **777.1**/231.4 | 15.39 | 67.30 | 40.33 | 75.2/79.2/68.8 |
| MM-LLAMA2-7B | | | | | |
| Finetune-80K | **661.3/237.1** | 16.09 | 65.08 | 31.64 | 56.3/65.0/55.4 |
| LayerNorm-80K | 583.2/200.7 | **16.78** | **88.85** | **49.24** | **66.6/68.5/64.9** |
| LayerNorm-Conv. 20K | 376.2/157.5 | 16.19 | 86.80 | 44.88 | 50.5/50.7/50.3 |
| MM-LLAMA2-CHAT-7B | | | | | |
| Finetune-80K | **805.4/234.6** | 15.29 | 57.40 | 26.70 | 60.3/69.8/57.9 |
| LayerNorm-80K | 651.3/219.3 | **16.60** | **75.34** | **43.75** | **71.3/72.4/67.8** |
| LayerNorm-Conv. 20K | 482.9/172.1 | 13.88 | 66.85 | 41.95 | 62.7/71.7/61.3 |
| MM-LLAMA2-13B | | | | | |
| Finetune-80K | 402.3/199.3 | **18.33** | 73.88 | 45.33 | 51.6/51.1/52.2 |
| LayerNorm-80K | 526.0/177.5 | 15.31 | **82.92** | **48.42** | 60.0/69.1/58.9 |
| LayerNorm-Conv. 20K | **646.0/242.9** | 16.01 | 76.50 | 44.86 | **70.0/76.9/68.6** |
| MM-LLAMA2-CHAT-13B | | | | | |
| Finetune-80K | 623.3/221.4 | 15.17 | 64.19 | 41.82 | 67.6/64.8/64.5 |
| LayerNorm-80K | **929.3/254.3** | **16.10** | **74.96** | 42.79 | **78.9/83.9/74.3** |
| LayerNorm-Conv. 20K | 769.7/227.5 | 15.57 | 73.30 | **43.08** | 68.2/72.8/65.3 |

Table 4: Results of models with LayerNorm and/or vision-language Connector activated.

| Method | MME | VQAv2 | MSCOCO | Flickr30k | POPE |
|--------|-----|-------|--------|-----------|------|
| MM-LLAMA2-7B | | | | | |
| LayerNorm + Connector | **583.2/200.7** | 16.78 | **88.85** | **49.24** | 66.6/68.5/64.9 |
| Connector | 311.1/105.4 | 12.72 | 60.43 | 35.91 | **67.9/73.7/66.9** |
| LayerNorm | 395.0/191.4 | **18.18** | 80.13 | 41.68 | 50.3/51.3/50.2 |
| MM-LLAMA2-13B | | | | | |
| LayerNorm + Connector | **526.0**/177.5 | 15.31 | **82.92** | **48.42** | 60.0/**69.1**/58.9 |
| Connector | 507.0/187.9 | 15.22 | 62.60 | 25.13 | **60.9**/66.8/**60.1** |
| LayerNorm | 405.0/**188.6** | **16.51** | 70.41 | 39.86 | 50.9/52.7/51.0 |

## 5.1 LAYERNORM TUNING ADAPTS LLMs TO MULTI-MODAL

**Influence of the Vision-Language Connector** The vision-language connector serves as the converter to project features from the vision encoder to the LLM domain. In our previous experiments, we focused on finetuning the LLM component of the MLLMs while keeping the vision-language connector activated by default. To determine which component plays a more important role for domain adaptation of LLM to multi-modal domain, we performed an ablation study by activating the two components separately. Results are presented in table 4, tuning LayerNorm in attention blocks without activating the vision-language connector resulted in only a 4.2% and 5.4% decrease in performance on three traditional multi-modal tasks and the MME benchmark, respectively. This decrease is significantly lower than the 15.6% and 9.2% downgrade observed when only activating the Connector on the same tasks. This observation highlights the vital role LayerNorm plays in transforming knowledge from the vision domain to language, indicating LayerNorm as a strong domain adaptor for the LLM architecture.

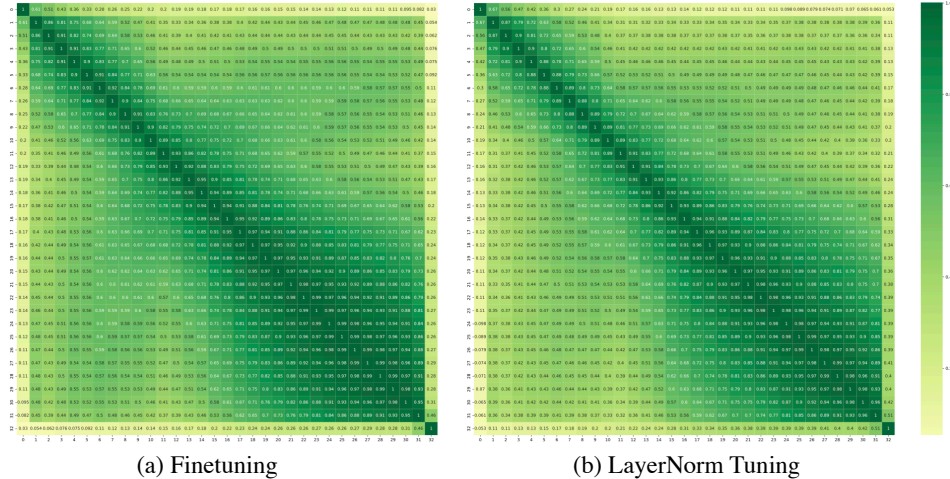

|                  |                  |
|:----------------:|:----------------:|
| (a) Finetuning   | (b) LayerNorm Tuning |

Figure 3: Layer similarities between different LLM layers in (a) Finetuned and (b) LayerNorm-tuned MM-VICUNA-7B. The average layer similarity of two models are 0.624 and 0.585, respectively.

Table 5: Results of models with LLAMA2 Finetuned/LayerNorm-tuned with ViT pre-trained on `ImageNet` (Deng et al., 2009), which have not been aligned with the language domain.

|                | MME            | VQAv2     | MSCOCO | Flickr30k | POPE            |
|----------------|----------------|-----------|--------|-----------|-----------------|
| Finetune-7B    | **406.79**/**182.5** | 15.05     | 47.75  | 18.97     | **50.0**/**51.6**/**50.1** |
| LayerNorm-7B   | 301.51/127.14  | **15.48** | **66.22** | **31.73** | **50.0**/50.1/**50.1** |
| Finetune-13B   | 375.41/**171.79** | **25.38** | 51.26  | 25.96     | 50.3/51.1/**51.0** |
| LayerNorm-13B  | **445.98**/150.0 | 15.59     | **64.63** | **32.17** | **51.2**/**53.0**/50.8 |

**Switching Visual Features.** We employ the ViT encoder from CLIP (Radford et al., 2021) by default in our previous experiments. CLIP (Radford et al., 2021) models are trained with image-text contrastive loss, thus its feature space is already aligned with language. Since LayerNorm has shown its effectiveness as a domain adaptor, we are interested in testing whether or not LayerNorm tuning can adapt a LLM to image features that are not pretrained to align with language. The vision encoder is switched to a ViT model that was pretrained on `ImageNet` (Dosovitskiy et al., 2021; Deng et al., 2009). Results in table 5 demonstrate that both LayerNorm and finetuning approaches can yield high performance. Interestingly, we observe that by LayerNorm tuning with `ImageNet` trained ViT, which has not been aligned with language, the model is able to achieve comparable performance to full parameter finetuning, *i.e.*, results show that LayerNorm tuning outperforms finetuning by 12.0% on captioning tasks, but performs slightly worse by 5.0% on the `MME` benchmark. These results again indicates the domain adaptor role of the LayerNorm , hinting the reason behind the empircal success of LayerNorm tuning. Furthermore, it is worth noting that the performance of MLLMs incorporating ViT pretrained on `ImageNet` is generally inferior to that of CLIP's vision encoder. This observation provides compelling evidence that, despite differences in tokenizer and training paradigm between CLIP's text encoder and LLAMA's, ViT from CLIP has the capacity to learn general patterns of language formulation during pre-training. Thus, significantly enhance MLLM abilities.

## 5.2 LAYERNORM TUNING IMPROVES THE EXPRESSIVE POWER

It is shown in Pires et al. (2023) that a Transformer model incorporating anisotropic layer representation can capture a wider range of learning patterns. By computing the cosine similarities between all layers in the LLM of a finetuned MLLM, we aim to investigate whether the improved efficiency is the results of the improved expressive power. In table 6, we present the average layer similarity of three 7B scale MLLMs, and in fig. 3 we present the visualization of per layer similarity scores of MM-VICUNA-7B. Our analysis reveals that the transformer layers in the MLLMs with LayerNorm tuning exhibit a clear distinction from one another (*i.e.*, an average 10.6% lower layer similarities comparing finetuning), indicating superior generalization ability and expressive power compared to

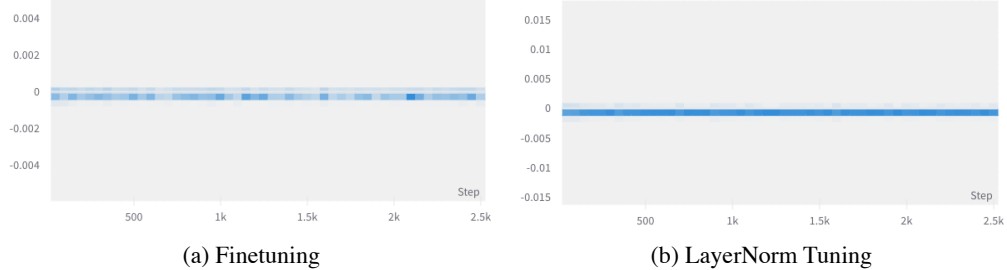

(a) Finetuning            (b) LayerNorm Tuning

Figure 4: Gradients of the input LayerNorm in the 11th layer of the MM-VICUNA as training proceeds. LayerNorm-tuned model has lower gradient variance than full parameter finetuning.

finetuning. This finding sheds light on why tuning LayerNorm is effective for multi-modal LLM training. For additional visualizations, please refer to the Appendix A.2.1.

Table 6: Layer representation similarity of LayerNorm and finetuning methods on three 7B MLLMs. Lower the similarity is, the better expressive power a model possesses.

| Model | LayerNorm Sim. | Finetuning Sim. |
|---|---|---|
| MM-VICUNA | 0.585 | 0.624 |
| MM-LLAMA2 | 0.504 | 0.591 |
| MM-LLAMA2-CHAT | 0.550 | 0.617 |

### 5.3 LAYERNORM TUNING HAS SMALLER GRADIENT VARIANCE

A well accepted view about LayerNorm is that, as the neural network goes deeper, the mean of LayerNorm gradients should goes to zero as the LayerNorm itself is designed to normalize all training parameters. In the meantime, the variance of LayerNorm gradients should be small to ensure a better generalization ability of the model (Xu et al., 2019) (See the proof in Appendix A.2.2). As we presented in fig. 4, MLLM with LayerNorm tuning method has a more concentrated LayerNorm gradients than fine-tuning during the training process. This result gives another view on the effectiveness of LayerNorm from the optimization perspective. More visualizations are listed in Appendix A.2.2.

## 6 CONCLUSION AND DISCUSSIONS

**LayerNorm is effective and sufficient built upon MLLM pre-training.** MLLM training typically involves pre-training on image-text pairs followed by finetuning on visual instruction data. While the second stage of training receives more attention, it is worth noting that the function of the first stage pre-training is non-negligible for training a competent MLLM. We have presented in the paper only a small portion of parameter activation is sufficient to tune a well-behaved MLLM. However, other models such as INSTRUCTBLIP (Dai et al., 2023) and MINIGPT4 (Zhu et al., 2023) only tune the vision-language connector, leaving the LLM untouched during the second stage of training. These models have yielded strong performances when given a large-scale finetuning dataset. In Sec. 5.1, we demonstrate that tuning LayerNorm may be a more effective means for the second stage training, especially when compared to existing parameter-efficient methods for training MLLMs.

**Limitations.** One shortcoming of these parameter-efficient finetuning methods is that they are more sensitive to hyper-parameters (*e.g.*, learning rate, training epoch) than finetuning. Since the number of trainable parameters of LayerNorm is small, the model performance of LayerNorm method also varies when twitching the training hyper-parameters. This drawback calls for potential future investigations on the LayerNorm tuning method. In the Appendix A.1, we give a hint for the grid search range of learning rate on both 7B and 13B scaled models using LayerNorm tuning based on our experimental results.

**Conclusion.** Our studies demonstrate LayerNorm tuning as a simple yet effective tuning method for adapting LLMs comprehend multi-modal content across various model variants. Compared to LoRA

tuning or full parameter finetuning, LayerNorm tuning reduces the trainable parameters by a significant 41.9%, enabling efficient finetuning of MLLMs on consumer-grade GPUs. Moreover, we demonstrate that MLLMs can achieve exceptional performance with minimal "right" data and parameters, showcasing the potential of LayerNorm tuning method in real-world applications. Given the empirical success of LayerNorm tuning, we revisited the MLLM finetuning from a domain adaptation perspective and showed that LayerNorm plays a critical role in adapting LLMs to the multi-modal domain. Additionally, our research illustrates the expressive power and optimization potential of LayerNorm tuning from layer similarities and the gradient variance. We hope that our work could inspire future works on designing improved PEFT methods that enable more diverse application scenarios for MLLMs.

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

## A APPENDIX

### A.1 TRAINING DETAILS

For the first stage, we set the learning rate to 2e-3 for all variants. During the second stage, we search learning the learning rate from [2e-3, 1e-3, 6e-4, 3e-4, 1e-4, 5e-5, 2e-5, 1e-5, 6e-6, 1e-6, 1e-7] for all models and pick the best learning rate based on their performances on the CIDEr score on the `Flickr30k` task.

According to our tryouts based on `Flickr30k` results in Table A1, the recommended learning rate for 7B scale is between 6e-4 to 2e-3, while on the 13B, the learning rate should be searched in the range of 3e-6 to 6e-5.

Table A1: Performance of MLLMs (LayerNorm-simp.) trained with different learning rates and scales on the `Flickr30k` task.

| Learning Rate | 3e-6 | 1e-5 | 3e-5 | 6e-5 |
|---|---|---|---|---|
| MM-LLAMA2 7B | 21.42 | 32.45 | **43.04** | 28.24 |
| Learning Rate | 6e-4 | 1e-3 | 2e-3 | - |
| MM-LLAMA2 13B | 37.35 | **46.88** | 44.15 | - |

### A.2 INSIGHTS OF LAYERNORM TUNING

#### A.2.1 VISUALIZATION EXAMPLES OF LAYER SIMILARITIES

Lower similarities between different layers of the transformer indicates more expressive power (Pires et al., 2023). In section 5.2, we have shown the computed cosine similarity between layers on a Vicuna model, here we show the layer similarities between layers on LLAMA2 and LLAMA2 CHAT models in fig. A1 and fig. A2. It is clear that, LayerNorm tuning again allows the model to learn dissimilar layer representations, improving the expressive power of the model.

#### A.2.2 GRADIENTS OF LAYERNORM

**Visualization examples of LayerNorm gradients.** In fig. A3 and fig. A4, we present the gradients of the LayerNorm parameters during the training process. Similar to the one we have shown in the main text, LayerNorm tuning demonstrates a smaller gradient variance which is important for converging to a better local minimum (Xu et al., 2019).

**Proof of smaller variance in LayerNorm .** As stated in Sec. 5.3, deeper the network is, the variance of LayerNorm in the model should be naturally smaller (Xu et al., 2019). We first let $\mathbf{y} = (y_1, y_2, ..., y_N)$ be the normalized vector, meaning the mean and variance of $\mathbf{y}$ is 0 and 1, respectively. We can then formulate the standard LayerNorm as follow:

$$\mathbf{y} = \frac{\mathbf{x} - \mu}{\sigma}, \quad \mu = \frac{1}{N} \sum_{i=1}^{N} x_i, \quad \sigma = \sqrt{\frac{1}{N} \sum_{i=1}^{N} (x_i - \mu)^2}, \tag{1}$$

where $\mathbf{x} = (x_1, x_2, ..., x_N)$ is the input vector and $N$ is the dimension of $\mathbf{x}$. $\mu$ and $\sigma$ are the mean and standard deviation of $\mathbf{x}$.

We first define $\mathbf{1}_N = \underbrace{(1, 1, ..., 1)^\mathsf{T}}_{N}$. For calculating the gradients of the normalized vector $\mathbf{y}$, we first simulate the backward propagation regarding the loss $\ell$:

$$\frac{\partial \ell}{\partial \mathbf{x}} = \left( \frac{\partial \mathbf{y}}{\partial \mathbf{x}} + \frac{\partial \mu}{\partial \mathbf{x}} \frac{\partial \mathbf{y}}{\partial \mu} + \frac{\partial \sigma}{\partial \mathbf{x}} \frac{\partial \mathbf{y}}{\partial \sigma} \right) \frac{\partial \ell}{\partial \mathbf{y}} = \frac{1}{\sigma} \left( I - \frac{\mathbf{y}\mathbf{y}^\mathsf{T}}{N} - \frac{\mathbf{1}_N \mathbf{1}_N^\mathsf{T}}{N} \right) \frac{\partial \ell}{\partial \mathbf{y}}. \tag{2}$$

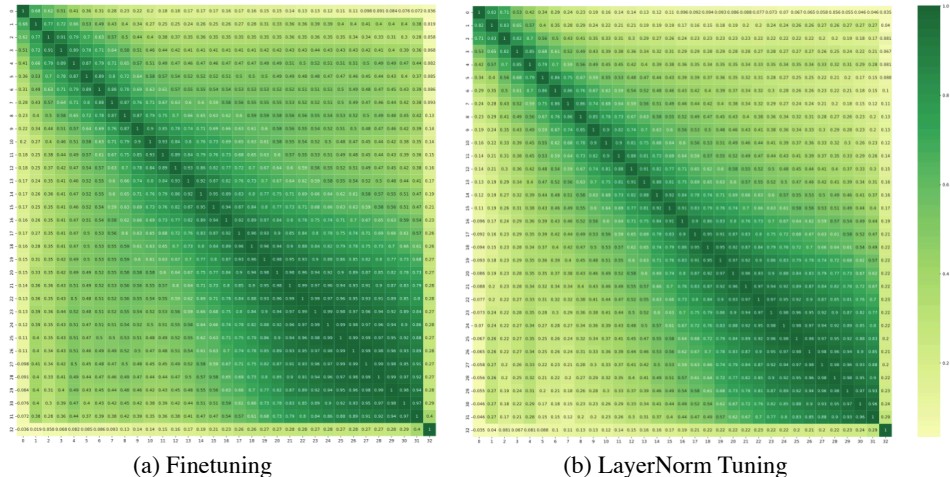

|  | (a) Finetuning |  | (b) LayerNorm Tuning |  |

Figure A1: Layer similarities between different LLM layers in (a) Finetuned and (b) LayerNorm-tuned MM-LLAMA2-7B.

Here we define $\frac{\partial \ell}{\partial \mathbf{x}} = (a_1, a_2, ..., a_N)$ with mean $\bar{a}$ and standard deviation $D_a$, and $\frac{\partial \ell}{\partial \mathbf{y}} = (b_1, b_2, ..., b_N)$ with mean $\bar{b}$ and standard deviation $D_b$. We set $W_1 = I - \frac{\mathbf{y}\mathbf{y}^\intercal}{N} - \frac{\mathbf{1}_N\mathbf{1}_N^\intercal}{N}$, we can verify that:

$$\mathbf{1}_N^\intercal W_1 = \mathbf{1}_N^\intercal \frac{1}{\sigma}\left(I - \frac{\mathbf{1}_N\mathbf{1}_N^\intercal + \mathbf{y}\mathbf{y}^\intercal}{N}\right) = \frac{1}{\sigma}\left(\mathbf{1}_N - \frac{\mathbf{1}_N^\intercal\mathbf{1}_N}{N}\mathbf{1}_N^\intercal - \frac{\mathbf{1}_N^\intercal\mathbf{y}}{N}\mathbf{y}^\intercal\right) = \frac{\mathbf{1}_N - \mathbf{1}_N - 0}{\sigma} = 0 \tag{3}$$

Therefore, we can easily proof that $N\bar{a} \propto \mathbf{1}_N^\intercal W_1 \bar{b} = 0$, which means the mean of $\frac{\partial \ell}{\partial \mathbf{x}}$ should be zero.

Then we dive into proofing the variance of LayerNorm gradients should be small when the number of network parameters $N$ becomes large.

$$\begin{aligned} D_a &= \sum_{i=1}^{N}(a_i - \bar{a})^2/N = \sum_{i=1}^{N}a_i^2/N \\ &= \left\|(a_1, a_2, \ldots, a_N)^\intercal\right\|^2/N \\ &= \left\|W_1(b_1, b_2, \ldots, b_N)^\intercal\right\|^2/N \\ &= \left\|W_1(b_1 - \bar{b}, b_2 - \bar{b}, \ldots, b_N - \bar{b})^\intercal + W_1\bar{b}\mathbf{1}_N\right\|^2/N \\ &= \left\|W_1(g_1 - \bar{b}, g_2 - \bar{b}, \ldots, g_N - \bar{b})^\intercal\right\|^2/N \\ &\leq W_1^2 \sum_{i=1}^{N}(b_i - \bar{b})^2/N \end{aligned} \tag{4}$$

Since the projection matrix $W_1$ is idempotent, we have $W_1^2 = W_1$. That is to say, when $N$ is large enough, there stands $D_a \leq (I - \frac{\mathbf{y}\mathbf{y}^\intercal + \mathbf{1}_N\mathbf{1}_N^\intercal}{N})\sum_{i=1}^{N}(b_i - \bar{b})^2/N \propto 1/N^2$. As a consequence, when the network parameter $N$ is large, the gradient variance of LayerNorm should be small.

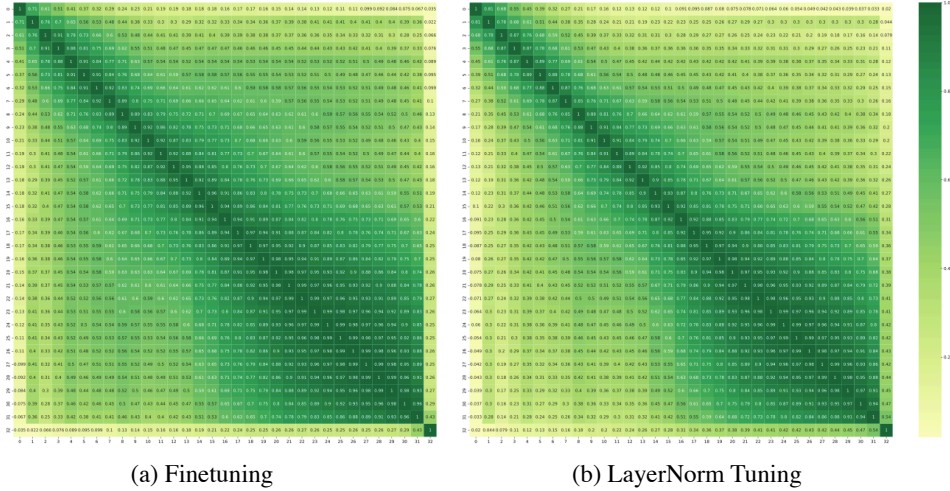

(a) Finetuning

(b) LayerNorm Tuning

Figure A2: Layer similarities between different LLM layers in (a) Finetuned and (b) LayerNorm-tuned MM-LLAMA2-7B CHAT.

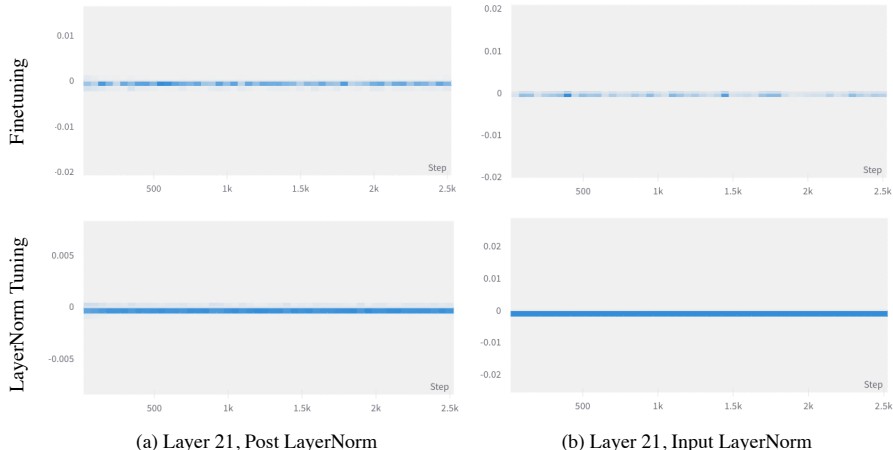

(a) Layer 21, Post LayerNorm

(b) Layer 21, Input LayerNorm

Figure A3: The gradients of both input and post LayerNorm in 21st layer of the MM-VICUNA as the training proceeds.

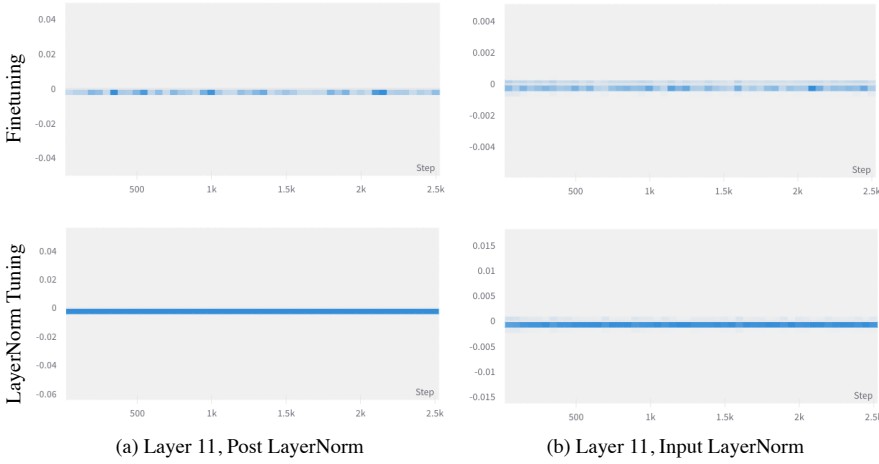

(a) Layer 11, Post LayerNorm

(b) Layer 11, Input LayerNorm

Figure A4: The gradients of both input and post LayerNorm in 11th layer of the MM-VICUNA as the training proceeds.

