# OpenReview forum: "Tuning LayerNorm in Attention: Towards Efficient Multi-Modal LLM Finetuning"
_ICLR.cc/2024/Conference — ICLR 2024 spotlight_

### Official Review · Reviewer_6WJ1 · 2023-10-27

**Soundness:** 3 good
**Presentation:** 3 good
**Contribution:** 3 good
**Rating:** 6
**Confidence:** 4

**Summary:**

This paper proposes an interesting idea of tuning LayerNorm for PEFT, yielding impressive results compared with LoRA and tuning other parts in LLMs. Five multi-modal tasks were tested to show the effectiveness of their proposed method. 20% better performances, 41.9% reduction of trainable parameters and 17.6% decreasing of GPU memory usage were reported. The contributions of this paper include, (1) tuning LayerNorm with a simple and efficient direction, and (2) significantly better results and resource/cost reduction on 5 tasks.

**Strengths:**

1. simple and efficient tuning of LayerNorm for LLMs;
2. strong results reported for 5 multi-modal tasks.

**Weaknesses:**

1. simple layernorm finetuning is reported to be including less trainable parameters and sensitive to a list of hyperparameters such as learning rate and training epoch, making it difficult to find the optimal performances in a quick way;
2. not clear yet of how other researchers combine LayerNorm with other PEFT methods - are the results better or not?

**Questions:**

1. any further detailed results on combining LayerNorm tuning with other types of PEFT? such as those listed in https://github.com/huggingface/peft?
2. so how exactly do you select the hypermeters during tuning? learning rate and training epoch, any detailed hints on applying your method to new tasks and new datasets?
3. can you show some examples that tent to be better or worse when applying LayerNorm?

---

> ### Author Response · Authors · 2023-11-19
>
> >**Questions 1 and Weakness 2**: Results of combining LayerNorm tuning with other PEFT.
>
> We have provided the results of combining LayerNorm tuning with other PEFT in the response to Reviewer qMBg. The results of combined methods show an improvement from using only one of the tuning methods, indicating that LayerNorm is orthogonal to previous PEFT methods.
>
> >**Questions 2 and Weakness 1**: How to select the hyperparameters.
>
> We have presented the parameter search procedure in Appendix A.1, where we searched for a range of learning rates for training the model. For the training epoch, we follow the standard practice of LLaVA to only train for **1 epoch** for both the pretraining stage and the fine-tuning stage **for all tuning methods**.
>
> For applying LayerNorm tuning for new datasets and tasks, we recommend setting the learning rate to a higher value, as in our experiments, for LayerNorm tuning to perform well, the learning rate typically need to be set to 10x larger than LoRA tuning or full fine-tuning. We also present our explorations w.r.t. learning rate in the Appendix A.1, and **give our recommended searching range for both 7B and 13B models**.
> We will include more detailed discussions in the final version of our paper.
>
>
> >**Questions 3**: Cases of success and failure of applying LayerNorm Tuning.
>
> In our paper, we have used the MME benchmark which contains the evaluation of MLLMs on both cognition and perception aspects, as shown in our Table 1 of the main paper. We can see that LayerNorm tuning can bring consistent improvements or at least match other tuning methods on the perception aspects. On the cognition aspect, the improvement is not consistent, and sometimes LayerNorm tuning falls behind other methods like LoRA tuning. However, note that LayerNorm tuning activates 10x less parameters than LoRA tuning, the performance gain from it is considered to be impressive.
> We will include these discussions in the final version of our paper.

---

> ### Author Response · Authors · 2023-11-21
>
> Dear Reviewer 6WJ1:
>
> We want to thank you here, again, for the constructive comments and the positive acknowledgment of this paper. We have provided explanations to try to address your concerns. Could you please kindly check our responses, to see if your concerns are solved? We would really like to hear if you have any further questions before the discussion window is over. And if no more questions, please could you consider updating the score?
>
> Sincerely,
>
> Authors

---

### Official Review · Reviewer_itTB · 2023-10-31

**Soundness:** 3 good
**Presentation:** 4 excellent
**Contribution:** 3 good
**Rating:** 6
**Confidence:** 3

**Summary:**

The paper introduces an efficient approach to finetuning Large Language Models (LLMs) for multi-modal tasks, resulting in Multi-modal Large Language Models (MLLMs). By finetuning only the LayerNorm's weights within attention blocks and making LayerNorm the sole trainable component, the model achieves superior performance, especially when finetuned with conversational data. This LayerNorm-focused strategy, combined with the right data type, leads to significant performance improvements on benchmarks while using fewer resources, highlighting the potential of LayerNorm tuning for multi-modal tasks. The experiments are thorough and well-executed, yielding insightful results.

**Strengths:**

1. The paper introduces an efficient way for transforming LLMs into MLLMs by only finetuning the LayerNorm within each attention block.
2. The paper provides empirical results showing that the proposed LayerNorm tuning method can yield competitive performance, often surpassing other tuning approaches like full parameter finetuning or LoRA.
3. The LayerNorm tuning method offers substantial benefits in terms of efficiency, reducing trainable parameters and GPU memory usage.
4. Beyond empirical results, the paper explores deep into understanding the role of LayerNorm in adapting LLMs to the multi-modal domain, enhancing the model's expressive power.

**Weaknesses:**

1. Inconsistency Across Different LLMs: The experimental results appear to vary depending on the specific LLM used. While the authors highlight that human-aligned LLMs tend to yield better results, there seems to be inconsistency, especially when comparing the human-aligned 7B model with the 13B one. This raises questions about the robustness of the proposed method across different model sizes and configurations, making the claim less convincing.
2. Lack of Exploration of Combined Training Strategies: The paper primarily focuses on the LayerNorm tuning strategy. A more comprehensive exploration, combining this with other training strategies, would provide a richer understanding of the method's potential and its interaction with other techniques. Such an exploration could offer more actionable insights for practitioners looking to adopt the proposed method.
3. Empirical and Intuitive Explanations: The paper's explanations for the observed results lean heavily on empirical evidence and intuitive reasoning. A deeper theoretical analysis or a more rigorous exploration of the underlying mechanisms would strengthen the paper's claims and provide a more solid foundation for the proposed method.

**Questions:**

1. In the experiments, there seems to be a performance variance between different LLMs, especially between the human-aligned 7B and 13B models. Could the authors elaborate on the potential reasons for this inconsistency?
2. While the paper provides empirical and intuitive explanations, could the authors shed light on any theoretical insights or hypotheses they might have about why tuning only the LayerNorm yields such significant results?
3. Beyond LoRA, are there other recent advancements or techniques that have shown promise in multi-modal learning or the finetuning of Large Language Models? It would be beneficial to compare your approach with these newer baselines to provide a more current context.

---

> ### Author Response · Authors · 2023-11-19
>
> >**Questions 1 and Weakness 1**: Inconsistency across LLMs.
>
> Comparing MM-LLaMA2-Chat-7B to MM-LLaAM2-Chat-13B, we can see that LayerNorm tuning on the 7B model gets a low score on the MME benchmark, but on the 13B model LayerNorm gets the best score on MME. Based on our analysis on the gradient variance of LayerNorm, we found that the larger the number of parameters, the smoother the gradient of LayerNorm will be, thus enabling a better optimization. The 13B model indeed contains more parameters than the 7B model, thus contributing to the better optimization of LayerNorm tuning, giving a better performance. Additionally, we would like to note that even at the 7B scale, our proposed LayerNorm tuning still gets the best performance on the VQAv2, MSCOCO, Flickr30k, and POPE benchmarks.
>
>
> >**Questions 2 and Weakness 3**: Theoretical insights of LayerNorm tuning.
>
> Our intuitions behind the success of LayerNorm tuning is provided in section 5, including the perspectives of domain adaptation, expressive power, and gradient variance. Apart from the empirical findings on the gradient variance in section 5.3, we have also provided theoretical proof in appendix A.2.2 showing that when the number of network parameters is larger as in the case of MLLMs, the gradient variance of LayerNorm will be smaller, providing a smoother optimization of the LayerNorm parameters. Thus giving such a significant result for parameter-efficient fine-tuning.
>
> >**Questions 3 and Weakness 2**: Compare and combine with more PEFT baselines.
>
> Thanks for the suggestion! We have performed the experiments of combining LayerNorm tuning with LoRA and LayerNorm tuning with Prefix-tuning. The results of MLLM with LLaMA-Chat-7B are presented in the table below.
>
>
> |                           | MME/CS | MME/PS | MS-COCO |
> |---------------------------|--------|--------|---------|
> | LayerNorm tuning          | 651.3  | 219.3  | 75.34   |
> | LoRA                      | 709.8  | 228.6  | 57.26   |
> | Prefix                    | 541.0  | 202.1  | 50.78   |
> | LayerNorm tuning + LoRA   | 734.8  | 245.6  | 75.61   |
> | LayerNorm tuning + Prefix | 607.2  | 229.1  | 76.6    |
>
>
> We can see that combining **LayerNorm tuning with other PEFT method brings additional improvement** in performance, indicating that the gain from LayerNorm tuning is orthogonal to the gains of other PEFT methods.
> We will include these experiments in the final version of the paper.

---

> ### Author Response · Authors · 2023-11-21
>
> Dear Reviewer itTB:
>
> We want to thank you here, again, for the constructive comments and the positive acknowledgment of this paper. We have provided explanations to try to address your concerns. Could you please kindly check our responses, to see if your concerns are solved? We would really like to hear if you have any further questions before the discussion window is over. And if no more questions, please could you consider updating the score?
>
> Sincerely,
>
> Authors

---

### Official Review · Reviewer_qMBg · 2023-11-01

**Soundness:** 3 good
**Presentation:** 3 good
**Contribution:** 3 good
**Rating:** 8
**Confidence:** 2

**Summary:**

This paper introduces a strategy for transforming Large Language Models (LLMs) into Multi-Modal Large Language Models (MMLMs) by tuning LayerNorm in attention. The authors argue that this approach can improve the performance and efficiency of multi-modal tasks, such as image captioning and visual question answering. The paper presents empirical evidence to support this claim, showing that the proposed method outperforms existing approaches on several benchmark datasets. The authors also discuss the role of LayerNorm in adapting LLMs to the multi-modal domain and improving the expressive power of the model. Overall, the paper provides a novel and effective solution to the challenge of multi-modal learning, which has important implications for natural language processing and computer vision.

**Strengths:**

1. This paper introduces a novel method to transform large language models into multi-modal language models by tuning LayerNorm only, which yields strong performance. The idea is straightforward but effective.
2. The experiments are solid and the results support the idea of the authors.
3. The paper is well-written and well-organized, and the comprehensive analysis is interesting and attractive.

**Weaknesses:**

1. It is better to detail the LayerNorm tuning way in the paper and compare it with other methods.
2. Although existing experiments have demonstrated the effectiveness of the method, I suggest evaluating the model's capabilities comprehensively on more data, such as VSR, HM...

**Questions:**

See above.

---

> ### Author Response · Authors · 2023-11-19
>
> >**Weakness 1**: Detail of LayerNorm tuning and comparison to other methods.
>
> The LayerNorm tuning methods have been detailed in Figure 1, where the parameters of the LayerNorm module, the vision-language connector, embedding module, and the output head is optimized when tuning MLLMs.
> The LayerNorm-simp. tuning method only activates the parameters of the LayerNorm module for tuning MLLMs.
> LayerNorm tuning requires the least number of parameters while achieving similar or better performances compared to other parameter-efficient methods like LoRA, which adds a new low-rank matrix alongside the QV projector in attention and MLP modules, and methods that activate the QV projector in attention (Attn QV proj.) and the MLP layers (Attn MLP).
> This discussion has been included in the section 3 of the paper (paragraph ‘Tuning Modules’)
>
> >**Weakness 2**: Experiments on more data can be performed.
>
> We agree with the reviewer that more evaluations could provide a more holistic understanding of the effectiveness of LayerNorm tuning. However, in this paper, we focus more on the general abilities of MLLMs, thus we choose to evaluate the model on MME which is considered as a comprehensive evaluation benchmark for MLLMs. We have also provided the results of evaluating LayerNorm tuning on MMLU and TruthfulQA in the response to reviewer rJin to demonstrate the effectiveness of LayerNorm tuning.
> We were not sure what VSR and HM stands for, we would gladly add the test on VSR and HM if the reviewer could point us to the resources for them.

---

> ### Author Response · Authors · 2023-11-21
>
> Dear Reviewer qMBg:
>
> We want to thank you here, again, for the constructive comments and the positive acknowledgment of this paper.
> We have provided explanations to try to address your concerns. Could you please kindly check our responses, to see if your concerns are solved?
> We would really like to hear if you have any further questions before the discussion window is over. And if no more questions, please could you consider updating the score?
>
> Sincerely,
>
> Authors

---

### Official Review · Reviewer_rJin · 2023-11-03

**Soundness:** 3 good
**Presentation:** 3 good
**Contribution:** 3 good
**Rating:** 8
**Confidence:** 4

**Summary:**

Transforming LLMs, to Multi-Modal LLMs requires further finetuning on multi-modal data which can be pretty expensive.

There exist parameter efficient alternatives to fully finetuning models, and most popular ones are prefix tuning and LoRA. These methods a small amount of trainable parameters to the model.

The paper presents an alternative, which is not only parameter efficient, but also doesn't add new parameters to the model. The method simply trains the layer norm weights of the model associated with attention. The authors also demonstrate through multiple experiments that this is a strong alternatives that perform pretty well for multi-modal setup. There are two alternatives discussed: 1) only LayerNorm, and 2) LayerNorm along with vision language connector, word embeddings and output head. While #1 is more efficient, #2 is more performant.

**Strengths:**

- Strong results while being much more parameter efficient compared to other similar techniques like LoRA, Attn QV Proj, and Attn MLP.
- The two variants present a tradeoff between parameter efficiency and quality. This could also be useful when the finetuning dataset is very small and using LayerNorm only variant could provide a less overfitting alternative.
- While LoRA and other methods add additional parameters (adapters), this method does not need it and hence does not require handling them separately during training/inference to merge them with the original parameters.

**Weaknesses:**

- The method is claimed to be useful only for MultiModal tuning, and might not be comparable to LoRA or other methods for LLM only finetuning.

**Questions:**

Do we have any results comparing it with LoRA for non-multimodal tasks?

---

> ### Author Response · Authors · 2023-11-19
>
> We would like to thank the reviewer for their insightful and positive comment.
>
> >**Questions 1 and Weakness 1**: Results on non-multi-modal tasks.
>
> We have performed experiment of LayerNorm tuning on the Platypus instruction tuning dataset (https://huggingface.co/datasets/garage-bAInd/Open-Platypus) and evaluation on MMLU and TruthfulQA which are both language only tasks.
> The results are shown as follows, we use the LLaMA-2-7B-chat models for the experiments.
>
> |                  | MMLU | TruthfulQA-mc1 |
> |------------------|------|----------------|
> | Fine-tuning      | 54.8 | 29.8           |
> | LoRA             | 56.7 | 30.4           |
> | LayerNorm tuning | 57.1 | 31.0           |
>
> For LayerNorm tuning, we can see that it is **still effective on language-only tasks** compared to LoRA tuning and full fine-tuning, indicating the effectiveness of the proposed tuning paradigm.
>
> Note that, since the size of Playtypus may be small for tuning a 7B model (*i.e.*, 25K training data, which is much smaller than the 80K multimodal data we employed for visual instruction tuning). The LoRA tuning is expected to be better than fine-tuning for the generation task under this situation [1].
>
> [1] Revisiting Parameter-Efficient Tuning: Are We Really There Yet? EMNLP 2022

---

> ### Author Response · Authors · 2023-11-21
>
> Dear Reviewer rJin:
>
> We want to thank you here, again, for the constructive comments and the positive acknowledgment of this paper.
> We have conducted additional experiments and provided explanations to try to address your concerns. Could you please kindly check our responses, to see if your concerns are solved?
> We would really like to hear if you have any further questions before the discussion window is over. And if no more questions, please could you consider updating the score?
>
> Sincerely,
>
> Authors

---

### Author Response · Authors · 2023-11-19
**Overall Response**

We appreciate all reviewers for the unanimously positive comments and constructive suggestions.
We are excited to see that our work is considered to be, “strong results” (rJin, 6WJ1), “does not need additional parameters”(rJin), “straightforward but effective” (qMBg), “solid” (qMBg), “offers substantial benefits in terms of efficiency”(itTB), “explores deep into understanding” (itTB), “simple and efficient” (6WJ1).

Additionally, we would like to highlight our insightful findings again: converting LLM to MLLM is a domain adaptation process, through LayerNorm tuning, **we achieve a significant 42% parameter reduction and a 20% performance boost** compared to LoRA.

Individual concerns have been addressed carefully in the response to each reviewer. In the final version, we will revise the paper following the suggestions.

---

### Meta-Review · Area_Chair_ddqV · 2023-12-03

**Metareview:**

The paper introduces an efficient approach to fine-tuning text-only Large Language Models (LLMs) for multi-modal tasks to reduce the expense involved in fine-tuning. By fine-tuning only the LayerNorm's weights within attention blocks and making LayerNorm the sole trainable component, the model achieves superior performance (e.g. compared to prefix tuning and LoRA), especially when fine-tuned with conversational data. This LayerNorm-focused strategy, combined with the right data type, leads to significant performance improvements on benchmarks while using fewer resources, highlighting the potential of LayerNorm tuning for multi-modal tasks using to variants: 1) only LayerNorm, and 2) LayerNorm along with vision language connector, word embeddings and output head. While #1 is more efficient, #2 is more performant.

Strengths
- This paper introduces a novel method to transform large language models into multi-modal language models by tuning LayerNorm only, which yields strong performance. The idea is straightforward but effective.
- The experiments are solid and the results support the idea of the authors.
- The paper is well-written and well-organized, and the comprehensive analysis is interesting and attractive.

Weaknesses
- The experimental results appear to vary depending on the specific LLM used. This raises questions about the robustness of the proposed method across different model sizes and configurations, making the claim less convincing.
- Lack of Exploration of Combined Training Strategies: The paper primarily focuses on the LayerNorm tuning strategy. A more comprehensive exploration, combining this with other training strategies, would provide a richer understanding of the method's potential and its interaction with other techniques. Such an exploration could offer more actionable insights for practitioners looking to adopt the proposed method.
- Empirical and Intuitive Explanations only: The paper's explanations for the observed results lean heavily on empirical evidence and intuitive reasoning. A deeper theoretical analysis or a more rigorous exploration of the underlying mechanisms would strengthen the paper's claims and provide a more solid foundation for the proposed method.

In their responses, authors provide additional results and allay the reviewers' concerns to some extent.

**Justification For Why Not Higher Score:**

Providing additional experimental results on other data types  and model sizes (e.g. 7B vs 13B) and additional theoretical analysis would make the paper even stronger (and put it into "oral" territory)

**Justification For Why Not Lower Score:**

Reviewers agree on the strength of the results and the importance of the results to the field, the overall quality of this paper is higher than regular "poster" publications.

---

### Decision · Program_Chairs · 2024-01-16

Accept (spotlight)